# Integrated Placental Modelling of Histology with Gene Expression to Identify Functional Impact on Fetal Growth

**DOI:** 10.3390/cells12071093

**Published:** 2023-04-06

**Authors:** Hannah Ee Juen Yong, Katarzyna Maksym, Muhammad Ashraf Bin Yusoff, Esteban Salazar-Petres, Tatiana Nazarenko, Alexey Zaikin, Anna L. David, Sara L. Hillman, Amanda N. Sferruzzi-Perri

**Affiliations:** 1Centre for Trophoblast Research, Department of Physiology, Development and Neuroscience University of Cambridge, Cambridge CB2 3EG, UK; 2Singapore Institute for Clinical Sciences (SICS), Agency for Science, Technology and Research (A*STAR), 30 Medical Drive, Brenner Centre for Molecular Medicine, Singapore 117609, Singapore; 3Elizabeth Gareth Anderson Institute for Women’s Health, University College London, 84-86 Chenies Mews, London WC1E 6HU, UK; 4Fetal Medicine Unit Elizabeth Gareth Anderson Wing, University College Hospitals NHS Trust, 25 Grafton Way, London WC1E 6DB, UK; 5Department of Mathematics, University College London, London WC1E 6AE, UK; 6National Institute for Health Research University College London Hospitals Biomedical Research Centre, 149 Tottenham Court Road, London W1T 7DN, UK

**Keywords:** placenta, FGR, modelling, morphology, growth genes, transport

## Abstract

Fetal growth restriction (FGR) is a leading cause of perinatal morbidity and mortality. Altered placental formation and functional capacity are major contributors to FGR pathogenesis. Relating placental structure to function across the placenta in healthy and FGR pregnancies remains largely unexplored but could improve understanding of placental diseases. We investigated integration of these parameters spatially in the term human placenta using predictive modelling. Systematic sampling was able to overcome heterogeneity in placental morphological and molecular features. Defects in villous development, elevated fibrosis, and reduced expression of growth and functional marker genes (*IGF2, VEGA, SLC38A1*, and *SLC2A3*) were seen in age-matched term FGR versus healthy control placentas. Characteristic histopathological changes with specific accompanying molecular signatures could be integrated through computational modelling to predict if the placenta came from a healthy or FGR pregnancy. Our findings yield new insights into the spatial relationship between placental structure and function and the etiology of FGR.

## 1. Introduction

The human placenta is a highly specialized organ in pregnancy, whereby normal functioning is critical to fetal development and long-term health. By acting as a functional interface between the maternal and fetal circulation, it is responsible for maternal–fetal substrate exchange, it protects the fetus from immune rejection by the mother, and it secretes hormones, which maintain pregnancy and promote healthy fetal growth and development [1,2]. This complex organ is structurally heterogeneous, not only in terms of its broad spatial macroscopic characteristics with multiple cotyledons and vascular supply, but also at the microscopic level of the villous tree and trophoblast. Even in pregnancies that deliver at term without any apparent pathology, novel techniques such as micro-CT are now describing the large degree of heterogeneity of vascular density and branching in the placenta [3]. In addition, computational models have predicted that local heterogeneity in placental vascular structure can have major impacts on the resistance of the fetoplacental circulation and placental dysfunction [4]. Investigating the spatial relationships between changes in gene expression and structural differences in the placenta is important, so that we may better elucidate the potential functional effects of placental pathology in obstetric disease.

Fetal growth restriction (FGR) is a major cause of perinatal morbidity and mortality. The risk of in utero or neonatal death is especially high amongst growth-restricted fetuses and neonates who are commonly born preterm [5]. In some regions of the world, FGR and prematurity account for 80% of neonatal deaths [6]. FGR is diagnosed when a fetus fails to reach their genetically predefined growth potential and is often identified by fetoplacental ultrasound features (such as increased umbilical artery resistance to blood flow), and subsequently confirmed through placental histopathology [7]. According to international consensus, the gestational age at diagnosis is used to further subdivide FGR into early-onset, detected before 32 weeks of gestation, and late-onset [8]. Neonates born following FGR have an increased risk of developing health problems, including low blood sugars and poor temperature regulation and feeding, both in the immediate postnatal period and also in later adult life in the form of cardiovascular and metabolic disease [9,10]. Management of pregnancies affected by FGR can be challenging, as timing of delivery has to be judged well, to balance the hazards of preterm birth against the risks of irreversible damage secondary to intrauterine hypoxia and nutritional deficiency or even stillbirth [11].

The placenta of FGR pregnancies is characterized by reduced syncytiotrophoblast surface area, increased thickness of the exchange barrier formed by the trophoblast and fetal capillary endothelium, and an increase in placental apoptosis [12,13,14,15]. Whilst it is recognized that there are key mediators of fetal growth that are influenced by placental function including, but not limited to, transfer of gases [16], glucose [17,18], lipids [19], and amino acids [18,20], understanding the mechanisms through which placental histological changes might be affecting these pathways remains a challenge. Nuanced, integrated investigations will better delineate the underlying functional mechanisms related to the histological changes observed in the placenta. Experimental animal data have highlighted roles central to fetal growth outcomes that certain structural and functional changes play in the placenta [21,22]. However, data on how structure and molecular changes together impact on function and on each other, even in the form of nutrient transporter expression, are scarce or not available for the human placenta [23].

Better understanding of how placental structure relates to functional capacity is further optimized through robust placental sampling protocols. Methodological errors arise from differences in sampling and/or tissue processing, which has previously hampered the reproducibility and possible extrapolation of placental studies [24,25]. Systematic uniform random sampling involves random selection of the first sampling site, with subsequent sites dictated by a pre-made sampling interval. This method is recommended to ensure representative and unbiased sampling of the placenta [25,26]. It is a simple method, allowing for even coverage of the placenta and ensures that all sites can only be selected once [25]. In practice, however, systematic uniform random sampling has limitations. Firstly, its lack of bias may be compromised if the pre-determined sampling interval coincides with a natural pattern existent in the placenta [26]. Secondly, placental templates designed for systematic uniform random sampling assume all placentas to be roughly circular in shape, with central umbilical cord insertion. This assumption of a uniform shape and cord insertion is, however, incorrect. Although studies do show that mean placental chorionic shape at term is round, deviations in placental shape are associated with reduced placental efficiency [27]. In addition, the site of umbilical cord insertion appears to be related to placental function, with a marginal cord insertion near the outer boundary of the placenta being associated with a more asymmetric chorionic vessel structure [28], which has been associated with FGR, stillbirth, and neonatal death [29].

While analysis of single sites may miss important histopathological and functional changes, multiple-site collection poses analytical challenges, and methods may still fail to identify critical differences within placentas affected by pathology such as FGR. Statistical modelling allows for the exploration and integration of multiple data points from the same sample to provide more accurate interpretation of functional consequences. Moreover, the use of modelling to predict regional and disease-state differences has high translational potential, for example, providing an explanation for why the pregnancy was compromised and which baby may benefit the most from early health monitoring and/or postnatal intervention. With this study we therefore, firstly, sought to better understand if systematic uniform random sampling methods could be optimized to the challenges that FGR placentas pose. Secondly, we sought to characterize the degree of detailed histological and molecular heterogeneity between multiple samples in healthy and FGR placentas and to examine potential links between histological changes and the expression of genes involved in placental formation, transport, and transcriptional regulation, and their relationships with fetal growth/pregnancy outcome. Finally, in a proof-of-principle approach, we integrated these changes into a novel analysis model to better delineate the significance of the placenta in health and disease.

## 2. Materials and Methods

### 2.1. Placental Tissue Collection and Sampling

Subjects were recruited from University College London Hospital NHS Foundation Trust, London, UK with ethics approval from the South-Central Oxford A Research Ethics Committee (17/SC/0432). After written informed consent was obtained, term placental biopsies were collected from subjects diagnosed with late-onset FGR (n = 7, FGR detected after 32 weeks’ gestation and an estimated fetal weight and/or abdominal circumference below 3rd centile on population-based charts used clinically [30]), and from control subjects with appropriately grown fetuses (n = 9 birthweight >10th centile and <90th centile). Data on maternal pre-existing conditions, previous obstetric history, ultrasound examination, and pregnancy complications were collected at the time of recruitment.

Placentas were weighed and sampled within 30 min of delivery. Sampling sites were chosen to be representative of four central and four peripheral regions always relative to cord insertion (Figure 1A). In marginal cord insertions, the placenta was orientated with the cord anterior. Sampling then took place relative to the cord insertion as in other cases. This meant there was one site that had less tissue to sample from initially, in non-central cord insertions, but that ultimately the same amount of tissue was sampled, and that tissue selection was directly related to the cord in the same way as all other samples. For each placenta, 8 tissue biopsies (with maternal tissue and fetal chorionic plate removed) were snap frozen using liquid nitrogen or dry ice and stored at −80 °C. Neighboring sampling sites were then immersion-fixed in 4% paraformaldehyde.

### 2.2. RNA Extraction, cDNA Synthesis, and qPCR

Total RNA was extracted from n = 9 control and n = 6 FGR frozen placental tissues (n = 1 FGR sample was not taken due to >30 min delay in freezing this sample) using the RNeasy Fibrous Tissue Mini Kit (Qiagen, Hilden, Germany) following the manufacturer’s protocol. Briefly, approximately 2 mg of tissue was added to 300 µL of lysis buffer and homogenized using a bead-based technique. Following treatment with proteinase K and addition of 100% ethanol, the homogenate was transferred to the spin columns, washed with supplied buffers and centrifuged to remove contaminants. Purified RNA was then eluted into RNase-free water. Quality and concentration of extracted RNA were determined by Nanodrop (Thermo Fisher Scientific, Waltham, CA, USA).

Complementary DNA (cDNA) was synthesized using the Applied Biosystems High-Capacity cDNA Reverse Transcription Kit (Thermo Fisher Scientific) according to manufacturer’s instructions on a thermal cycler. Real-time PCR was performed in duplicate using TaqMan™ Universal Master Mix II, with UNG (Thermo Fisher Scientific) and inventoried TaqMan^®^ gene expression assay probes (Thermo Fisher Scientific) with either VIC or FAM fluorophores for 5 housekeeping genes (*18SrRNA*, *B2M*, *GAPDH*, *GUSB*, and *YWHAZ*) and 8 genes of interest (*HNF1A*, *IGF2*, *SLC2A1*, *SLC2A3*, *SLC38A1*, *SLC38A2*, *SLC38A4*, and *VEGFA*) (Table 1). Only stable housekeeping genes (*18S rRNA*, *B2M*, *GADPH*, and *YWHAZ*) that showed no statistically significant differences between cases and controls were included in the housekeeper geomean calculation for normalizing gene expression across samples. The relative expression of genes of interest was then calculated using the 2^−∆∆Ct^ method.

### 2.3. Histological Preparation

Following paraformaldehyde fixation, biopsies of 8 sites from each control (n = 9) and FGR (n = 7) placenta were embedded in paraffin using routine histological techniques and sectioned at 7 µm thickness. Sections were rehydrated using xylene and ethanol gradients, stained with hematoxylin and eosin, dehydrated with ethanol gradients and xylene, then mounted with DPX. Slides were then scanned using a Nanozoomer digital slide scanner (Hamamatsu Photonics, Shizuoka, Japan).

### 2.4. Placental Stereology

Placentas were analyzed blinded to the diagnosis of FGR. To perform stereological analysis in a manner similar to that performed previously [31], transparent lattices with test points, test lines, or test arcs were superimposed onto scanned images viewed under different magnifications on the NDP.view2 software (Hamamatsu Photonics). Volume densities of the intervillous space, stem villi, intermediate villi, terminal villi, syncytial knots, and fibrosis (Figure 1B,C) were estimated by point-counting and a lattice of equally spaced test points arranged 4 by 4 under 10× magnification in at least 13 fields of view for a minimum of 200 measurements per section. Volume densities of the trophoblast, stroma, and fetal capillaries (Figure 1D) within villi were estimated under a similar lattice under 40× magnification in 20 fields of view per sample. Absolute placental volumes were estimated by multiplying volume densities with placental weight (g). Arithmetic barrier thickness was assessed under 100× magnification in 20 fields of view per sample and a superimposed lattice with equally spaced straight test lines, which, at times, intersected fetal capillaries and villous trophoblast involved in exchange. The ‘measure’ tool within the software was then used to determine the shortest distance between a fetal capillary and the maternal blood space, where there was an intersection. Mean harmonic barrier thickness was calculated using the inverse of mean reciprocal of each raw arithmetic barrier thickness measurement. Thickness uniformity index, as a measure of the variability in thickness across the villous membrane, was obtained from the ratio of the mean arithmetic barrier thickness to the mean harmonic barrier thickness. Surface densities were approximated by counting chance intersections of fetal capillaries and villous trophoblast involved in exchange with superimposed test arc lines under 40× magnification. To derive surface areas, surface densities were multiplied by placental villous volume. The theoretical diffusion capacity was calculated using the total surface area for exchange (averaged surface area of fetal capillaries and villi) divided by the mean harmonic barrier thickness and multiplied by Krogh’s constant for oxygen diffusion. Specific diffusion capacity was then calculated by dividing the theoretical diffusion capacity by the infant birthweight.

### 2.5. Predictive Modelling

To evaluate the informativity of different placental sampling locations (only peripheral, only central, or both regions together) and different data types (only stereological data, only PCR data, or both types together), we studied all their possible combinations with analysis performed in R (version 4.0.2). To analyze several well-established multi-dimensional methods of data analysis, the machine-learning models (*xgbTree*, *glmnet*, and *nnet*) were implemented using caret package (version 6.0.90). All 27 models were trained with a default set of hyperparameters, cross-validated on 5 folds (for a better selection of internal algorithm parameters), with seed parameter set to 123. The performance of models was assessed with receiver operating characteristic (ROC) curves using area under the curve (AUC) with 95% confidence intervals (*pROC* package, version 1.18.0). Results of AUCs were presented in two ways:

All samples’ AUC—calculating AUC considering all samples as independent;

All patients’ AUC—calculating AUC on patients’ results only (if all samples belonging to one patient were correctly predicted, then we assumed that a prediction for this patient was correct, if the prediction was wrong in at least one sample of the sample’s patient, then it was set as wrong for this patient).

For illustration of interdependencies of features, which are presented in Appendix A, we selected the best pairs that showed significant differences between FGR and control groups using ANOVA test after applying FDR (false discovery rate) adjustment to test’s *p*-values (stats package, Version 4.0.2).

### 2.6. Statistical Analyses

Normality of data was assessed by D’Agostino–Pearson normality test. ROUT testing for outliers was used to identify any outliers, which were then excluded from the analysis. For comparisons between two groups, Student’s *t* tests and Mann–Whitney U tests were performed for parametric and non-parametric continuous data, respectively. Categorical data between FGR and controls were analyzed by a 2 × 2 contingency table with Fisher’s exact test. Two-way ANOVAs followed by Sidak post hoc test for multiple comparisons were used to analyze mRNA expression data by pathology and sampling location. Graphpad Prism 6.01 (Graphpad Software Inc., La Jolla, CA, USA) was used for statistical analyses. A *p*-value of <0.05 was considered statistically significant for clinical characteristics and mRNA expression analysis. To account for multiple testing in histological analyses, a false discovery rate correction at 5% was applied to determine statistical significance.

## 3. Results

### 3.1. Patient Characteristics

We matched FGR cases and controls for maternal age, booking BMI, mode of delivery, and infant sex (Table 2) with available expanded clinical data (Appendix A). Nonetheless, several significant differences in clinical characteristics were still observed. FGR cases were more likely to be non-White and delivered infants with lower birth and placental weights about a week earlier, although all delivered at term (≥37 weeks of gestation).

### 3.2. Stereological Analysis of Placental Morphology

Comparing stereological findings by sampling site within the FGR or control groups showed no significant differences between samples taken from the central and peripheral regions of the placenta (Appendix A). Hence, we averaged the data of the central and peripheral regions of each placenta to compare by pathology (Table 3). Taking into consideration a false discovery rate of 5% for multiple testing, FGR placentas had significantly lower volume densities of intermediate villi and syncytial knots and higher volume densities fibrosis, and capillaries. Once placental weight was accounted for, FGR cases showed decreased volumes of intervillous spaces, intermediate villi, terminal villi, syncytial knots, and trophoblast and stromal components, and increased volume of fibrosis compared to control placentas. There were no differences in any measure of barrier thickness, surface area, surface densities, or diffusion capacities between FGR and control placentas; except that FGR placentas had a decreased villi surface area.

### 3.3. Placental Expression of Growth Factors and Nutrient Transporters

We also evaluated the mRNA expression of selected growth and transcriptional factors involved in regulating fetal growth and key nutrient transporters for glucose and amino acids (Figure 2) that have been previously associated with restricted fetal growth and development in human and animal studies [32,33,34,35,36]. Placental *IGF2*, *SLC2A3*, and *SLC38A1* expression was significantly lower for FGR versus controls, regardless of sampling location (values were lower in samples taken from both the central and peripheral parts of the FGR compared to control placenta; Figure 2A,D,E). *VEGFA* expression was significantly lower in the peripheral region as compared to the central region of FGR placentas, while no statistical difference was identified in the central region between cases and controls (Figure 2B). Only *SLC2A1* expression was significantly increased in FGR placentas as compared to controls, independent of sampling location (Figure 2C). No differences in mRNA expression of and *SLC38A2* were observed between FGR and control placentas or by sampling location (Figure 2F). In controls, placental *HNF1A* expression was higher in the peripheral region as compared to the central region (Figure 2G), but no differences were identified for FGR.

### 3.4. Associations between Placental Morphology and Expression of Functional Genes

We next evaluated relationships between placental morphology and the expression of growth factor and nutrient transporter genes within the different sampling sites, and between control and FGR groups using Pearson’s r coefficient (Figure 3). In FGR placentas, several inverse correlations were identified between expression of *IGF2*, *VEGFA*, *SLC38A1*, and *SLC2A3* and the volume of terminal villi, trophoblast, and capillary volume predominantly in the peripheral region of FGR placentas. Inverse correlations with fibrosis volume were also observed for expression *IGF2*, *VEGFA*, *SLC38A1*, and *SLC2A3* in the central region of FGR placentas. Additionally, the relative expression of *IGF2*, *VEGFA*, *SLC38A1*, and *SLC2A3* genes was inversely correlated with barrier thickness uniformity index and both maternal blood spaces and fetal capillaries surface densities in the FGR placenta (central and peripheral sites), although no significant correlations were detected for the control placenta. Moreover, the expression of *IGF2*, *SLC38A1*, and *SLC2A3* genes negatively correlated with both theoretical and specific diffusion capacity in FGR, but not control placentas. In contrast, FGR expression of *HNF1A* showed a positive correlation with specific diffusion capacity in the peripheral region. Positive correlations between the expression of all genes measured and fibrosis and stromal volumes were found for the peripheral region of control placentas, but not FGR placentas. Positive relationships were also seen for expression of *VEGFA* and *HNF1A* with the stem villi volume in central sites of control placentas.

### 3.5. Predictive Modelling

To evaluate the information gained from different placental regions/sampling sites (only peripheral, only central, or both together) and data types (only stereological data, only qPCR data, or both together), we considered all their possible combinations using predictive modelling (construction of classifiers that would make it possible to distinguish patients in the control group from patients with FGR). As predictive modelling was carried out using three machine learning (ML) methods; *xgbTree*, *glmnet*, and *nnet*, this resulted in 27 models for consideration. The binary outcome of FGR was used as an outcome with a chosen set of features (only stereological data, only qPCR data, and both data types together) used as predictors. The predictive performance of each model was assessed using a leave-one-out cross-validation (LOOCV) scheme: the prediction was made for each patient (with all measurements) by excluding (withholding) it from the dataset, training the classifier on the remaining (independent) samples, and then generating predictions for the withheld samples using the trained model. Using predictions on the training subset (on each round of LOOCV procedures), the best threshold (corresponding to the best sum of sensitivity plus specificity) distinguishing controls from FGR cases was found and a binary result was calculated for each withheld sample (1 indicated FGR if the prediction was above the threshold, and 0 indicated control if it was below the threshold). Through all rounds, LOOCV-procedure binary results were collected for withheld samples and performance was assessed using these sets of predictions. Performance was estimated using areas under the ROC curve (AUCs) in two ways: “AUC for all samples” and “AUC for all patients”, as described in the Materials and Methods section. For each combination of “placenta part + type of measurement + ML model” two such AUC values were obtained. Full results of these analyses can be found in Table 4 and a complete table of errors for each patient can be found in Appendix A). For a simplified visualization of these results, we considered the distributions of results (AUCs) of different types of ML models on different types of data obtained in different parts of the placenta for all patients and for all their samples separately (Figure 3). This plot clearly shows that among the considered models, the best are those based on measurements performed on the peripheral parts of the placenta and using stereological and qPCR data together. As shown in Table 4, the best model was found to be *nnet*, i.e., the model using simple artificial neural networks for a classification. This model gave the best result for both “AUC for all samples” (0.979) and “AUC for all patients” (0.917). By considering the stereological and qPCR data separately, the stereological data showed a greater predictive power than the qPCR data (Figure 3, for all variants), which, may be due to the clearly different interdependences of the stereological parameters of the control group and the FGR group (the most significant interdependences are presented in Appendix A). However, a combination of stereology and qPCR data worked better than just stereology for analyses on peripheral parts of the placenta (Figure 4). The results obtained on the central parts of the placenta were lower than the results obtained on both parts of the placenta or only on the peripheral (boxplots for the central parts are slightly lower than the other options). Moreover, results using only peripheral parts were better than using a combination of both parts. Thus, the best model would be that based on measures made on the peripheral parts of the placenta using stereological and qPCR data together.

## 4. Discussion

There was no significant variability in the morphological features of the placenta within and between the peripheral and central sites of the placenta, regardless of whether they came from a healthy or FGR pregnancy. In addition, the findings indicate that alterations in placental morphology may be uniform across the placenta in late-onset FGR pregnancies without other complications, such as pre-eclampsia, that can lead to more variable histological changes across the placenta [37].

Despite established evidence that the placenta drives a number of the ‘great obstetrical’ syndromes, which include FGR, the relatively poor understanding of underlying placental mechanisms makes interpretation of how they contribute to clinical presentations difficult [38]. Heterogeneity within the placenta is thought to be responsible for some of the difficulties in directly relating disease state to placental findings. Our results suggest that the systematic sampling technique that we used was able to overcome potential morphological spatial heterogeneity of the term placenta, whether from normal or late FGR pregnancies. Indeed, peripheral versus central-region comparisons revealed unique features in gene-expression correlations and predictive-modelling results. These findings enable more clinical interpretations to be made of the results and indicate that use of this sampling technique may improve reproducibility between placental studies. This analysis suggests that at least one peripheral and central sample should be taken when assessing placental function, particularly in disease states.

Morphological differences between the control and FGR placenta were consistent with findings of others [39], and indicate villous maldevelopment, with fewer mature intermediate and terminal villi and reduced villous surface area, as well as elevated fibrosis. The increased villous capillary density and reduced syncytial knots in the studied FGR placentas are novel findings and may reflect adaptive responses to hypoxic-reoxygenation events secondary to reduced uteroplacental and/or fetoplacental flows in FGR pregnancies [40,41,42]. Moreover, these beneficial changes in placental morphology may be the explanation for the FGR babies in our study reaching near full term. Hence, morphological findings provide a clue to underlying placental mechanisms that may be subject to change, with the resultant amelioration of disease and improvement to fetal outcome, if they can be enacted in utero. There is already significant interest in delivery of targeted agents to the uterine arteries to improve blood flow [43]. Delivery of agents to target specific areas of pathology and functional deficit within the placenta to improve the clinical condition may also be a potential therapeutic option.

Our approach of combining morphological assessments with gene expression aimed to identify some key etiological pathways for the pathology seen. Expression of placental *IGF2*, *SLC2A3*, and *SLC38A1* was lower in samples taken from both the central and peripheral parts of the FGR placenta, and correlated with volumes and other specific placental features. These data suggest a widespread role for these molecules in placental structure and development in the presence of FGR. Indeed, *IGF2* is known to be important for the formation of the placental exchange interface in humans, among other species [36], and is expressed at lower levels in placentas showing FGR/small for gestational age in some [44,45], but not all, studies [46]. Prior work has shown that a genetic deficiency of *SLC2A3* in mice leads to FGR [35] which supports our findings, but is in contrast to other work reporting an upregulation of its encoded protein in the human placenta of late-onset FGR [47]. Of the system A amino acid transporters expressed by the human placenta, *SLC38A1* is key for system A activity at term [48]. Other work has also shown that placental system A activity is reduced in explants prepared from term FGR placentas [49]. *VEGFA* was significantly lower in the peripheral region of the placenta in FGR and correlated to the following placental pathology (terminal villi, trophoblast, and capillary volume) that might suggest a causal association. This is consistent with other work reporting lower placental *VEGFA* expression at term in late onset FGR [50,51], and the involvement of angiogenic factors more generally in normal and pathological pregnancies [52]. Our findings also have relevance for emerging pre-clinical research in placenta-directed gene therapy for FGR, including the utility of insulin-like growth factors and angiogenic regulators [53]. *HNF1A* was more highly expressed by the peripheral region compared with the central region in healthy, but not FGR, placentas. Little is known about the function of *HNF1A* in the placenta, although the human tissue atlas [54] indicates it is abundantly expressed by the villous syncytiotrophoblast [55]. Recent work has highlighted that *HNF1A* may regulate a large number of genes in trophoblast cells [56] and mediate metabolic changes [57], which could be important for the placental support of fetal growth, more broadly.

Using a modelling approach in our discovery cohort, we demonstrated the potential of predicting outcome (control or FGR) based on stereology and qPCR data. Stereology and qPCR measurements taken from the peripheral part of the placenta, and analyzed together, have the highest predictive power. To avoid any possible overfitting of the models, we did not perform any feature-selection in advance for tuning of the algorithm hyperparameters. Despite this, we were able to obtain predictions with very high accuracy, as, in fact, for the best model, only 1 sample out of 60 (4 samples from peripheral placenta part for each of 15 patients) considered was predicted incorrectly. Indeed, this subject was one that was most likely to be misclassified in all of the tested models. Nevertheless, we found that the morphological data clearly shows separation of features for control and FGR placentas and probably, this made it possible to obtain high quality constructed models. For the present study, we used a very strict LOOCV. Whilst modelling predictions were poor when qPCR data were used alone, adding these data to the analysis of stereology data improved prediction, hence confirming the importance of qPCR data and the underlying link of these data with functional features. Whilst our modelling approach shows potential, future work would benefit from utilizing a larger dataset, with extremes of pathology, to validate the model.

## 5. Conclusions

Overall, our data add to current understanding of placental function through comprehensive sampling of healthy placentas and those affected by disease (FGR) with careful deep histological analysis, integrated with expression of genes involved in key biological pathways regulating placental function. Whilst our modelling approach requires validation with further pathology analysis, it offers the promise of better diagnostic yield and a novel insight into biological pathways that affect fetal growth and pregnancy outcomes by employing multi-disciplinary approaches to deliver integrative mechanistic understanding.

## Figures and Tables

**Figure 1 cells-12-01093-f001:**
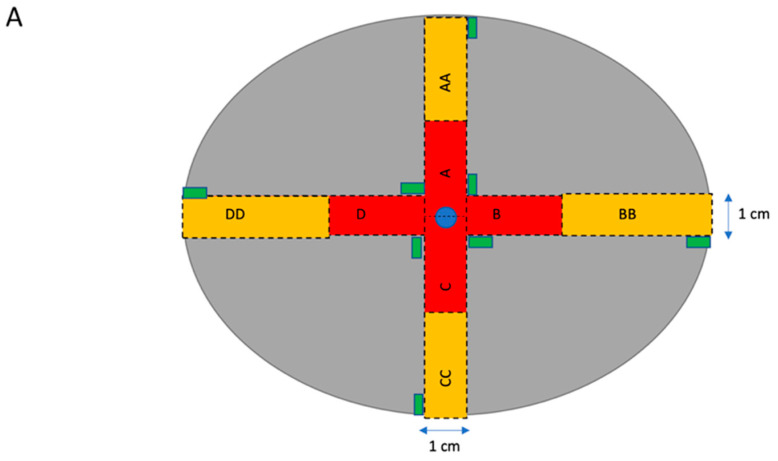
Diagram showing sampling of placenta (**A**) and main parameters measured in histological sections of placenta (**B**–**D**). (**A**) Dotted lines indicate where the placenta should be cut with a sterile histology knife. Red boxes with single letters indicate inner sections. Yellow boxes with double letters indicate outer sections. Green rectangles represent samples taken for RNA sampling. The central blue circle represents the umbilical cord (cut). (**B**–**D**) Representative histological sections of term human placenta. Stereological analysis was performed by identifying the intervillous space (IVS), stem villi (SV), intermediate villi (INTV), terminal villi (TV), fibrosis (FIB), and syncytial knots (*); and, under higher magnification of terminal villi, the trophoblast cell layer (#), fetal capillaries (+), and stromal cells (^). The scale bar denotes 50 µm.

**Figure 2 cells-12-01093-f002:**
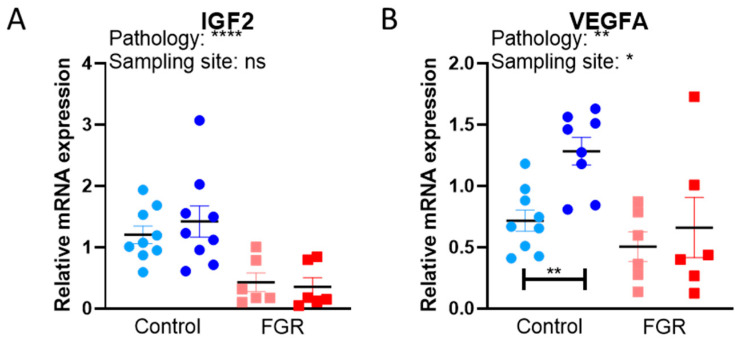
Relative mRNA expression of selected growth factors and nutrient transporters in control and FGR placentas. Expression of (**A**,**B**) growth factors (*IGF2* and *VEGFA*), and (**C**–**F**) transporters (*SLC2A1*, *SLC2A3*, *SLC38A1*, and *SLC38A2*) and the (**G**) transcription factor (*HNF1A*), were evaluated by qPCR. A total of three outliers were excluded; one control peripheral value for *VEGFA* and one control central value and one control peripheral value for *HNF1A*. Data are presented with individual data points with mean ± SEM and analyzed with two-way ANOVA, followed by Sidak’s post hoc test for multiple comparisons. * *p* < 0.05, ** *p* < 0.01, *** *p* < 0.001, **** *p* < 0.0001, ns = not significant.

**Figure 3 cells-12-01093-f003:**
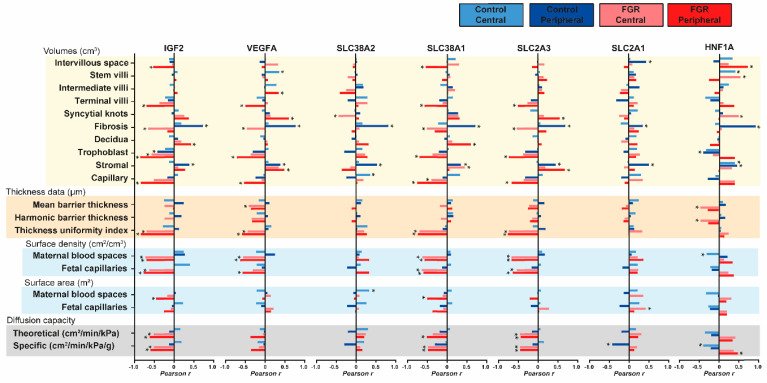
Correlations between stereological parameters and the relative expression of functional genes in the placenta. Pearson’s r coefficient was plotted and used to indicate strength of relationship between data (0: low, 1: high) and direction of correlation (+ sign: direct correlation and–sign: inverse correlation). Analysis was performed on each sampling site and experimental group (control and FGR) separately. Sample size was 36 for control and 24 for FGR. *: *p* < 0.05, correlation is statistically significant.

**Figure 4 cells-12-01093-f004:**
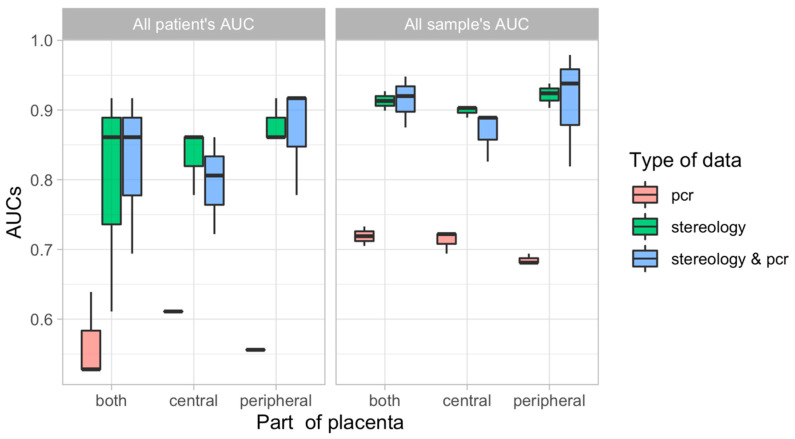
Distributions of AUCs, i.e., the performance results of different types of models on different types of data obtained on different parts of the placenta for all patients and for all their measurements separately.

**Table 1 cells-12-01093-t001:** Probes used for real-time PCR.

Analyzed Genes	Gene Symbol	Probe ID	Fluorophore
Housekeeping genes	*18S rRNA*	Hs99999901_s1	VIC
	*B2M*	Hs00187842_m1	VIC
	*GAPDH*	Hs02786624_g1	VIC
	*GUSB*	Hs00939627_m1	VIC
	*YWHAZ*	Hs01122445_g1	VIC
Genes of interest	*HNF1A*	Hs00167041_m1	VIC
	*IGF2*	Hs04188276_m1	FAM
	*SLC2A1*	Hs00892681_m1	FAM
	*SLC2A3*	Hs00359840_m1	FAM
	*SLC38A1*	Hs01562175_m1	FAM
	*SLC38A2*	Hs01089954_m1	VIC
	*SLC38A4*	Hs00394339_m1	FAM
	*VEGFA*	Hs00900055_m1	FAM

**Table 2 cells-12-01093-t002:** Patient characteristics.

Characteristics	Control (N = 9)	FGR (N = 7)	*p*-Value
Maternal age	34.44 ± 1.84	32.00 ± 1.16	0.339
Maternal ethnicity	8 white, 1 non-white	2 white, 5 non-white	0.035
Booking BMI (kg/m^2^)	24.67 ± 1.51	24.47 ± 2.75	0.946
Mode of delivery	4 vaginal deliveries, 5 caesarean sections	2 vaginal deliveries, 5 caesarean sections	0.620
Gestational age at delivery (weeks)	39.14 (39.07–39.79)	38.29 (37.29–39.00)	0.006
Infant sex	4F, 5M	2F, 5M	0.633
Infant birthweight (kg)	3.53 ± 0.13	2.33 ± 0.18	<0.001
Placental weight (g)	478.9 ± 16.6	340.7 ± 31.0	0.001

Data are presented as mean ± SEM and were analyzed by Student’s *t* test or Mann–Whitney U test. Categorical data were analyzed by 2 × 2 contingency table with Fisher’s exact test. BMI, body mass index; F, female; M, male.

**Table 3 cells-12-01093-t003:** Morphological differences in the placenta by pathology.

	Control (N = 9)	FGR (N = 7)	*p*-Value (^FDR)
**Volume density (cm^3^/g)**			
Intervillous space	0.37 ± 0.01	0.36 ± 0.02	0.447
Stem villi	0.06 ± 0.01	0.06 ± 0.01	0.895
Intermediate villi	0.12 ± 0.01	0.08 ± 0.01	0.008 *
Terminal villi	0.42 ± 0.02	0.4 ± 0.03	0.474
Syncytial knots	0.02 ± 0.00	0.01 ± 0.00	0.002 *
Fibrosis	0.01 ± 0.01	0.1 ± 0.02	0.001 *
Trophoblast	0.32 ± 0.01	0.34 ± 0.03	0.583
Stroma	0.55 ± 0.02	0.45 ± 0.05	0.058
Capillaries	0.13 ± 0.01	0.21 ± 0.03	0.016 *
**Volume (cm^3^)**			
Intervillous space	179.73 ± 9.45	119.44 ± 10.77	0.001 *
Stem villi	27.83 ± 3.2	20.17 ± 4.25	0.163
Intermediate villi	55.4 ± 4.41	25.48 ± 2.43	<0.001 *
Terminal villi	200.73 ± 10.54	135.3 ± 16.23	0.003 *
Syncytial knots	8.39 ± 0.78	2.44 ± 0.71	<0.001 *
Fibrosis	5.88 ± 3.24	37.02 ± 7.43	0.001 *
Trophoblast	153.5 ± 6.83	114.7 ± 13.69	0.017 *
Stroma	263.09 ± 15.46	153.23 ± 21.66	0.001 *
Capillaries	62.3 ± 5.04	72.78 ± 13.76	0.445
**Barrier thickness (µm)**			
Arithmetic mean	2.83 ± 0.08	2.85 ± 0.18	0.912
Harmonic mean	2.14 ± 0.07	2.24 ± 0.14	0.500
Thickness uniformity index (a measure of thickness variability)	1.32 ± 0.01	1.48 ± 0.08	0.044
**Surface density (cm^2^/cm^3^)**			
Villi	610.45 ± 23.85	630.28 ± 44.68	0.683
Fetal capillary	298.26 ± 19.81	401.92 ± 48.29	0.048
**Surface area (m^2^)**			
Villi	11.68 ± 2.88	10.59 ± 1.55	0.009 *
Fetal capillary	6.27 ± 1.7	6.89 ± 1.23	0.576
**Theoretical diffusion capacity (cm^2^/min/kPa)**	95.29 ± 4.86	70.26 ± 11.5	0.047
**Specific diffusion capacity (cm^2^/min/kPa/g)**	27.2 ± 1.49	29.47 ± 3.99	0.567

Data presented as mean ± SEM and analyzed by *t* test with FDR (false discovery correction) correction at 5%. * denotes significant findings that passed the 5% FDR.

**Table 4 cells-12-01093-t004:** Predictive modelling results using different machine-learning methods.

**Peripheral and Central**									
	**Stereology**			**qPCR**			**Stereology and qPCR**		
	**xgbTree**	**glmnet**	**nnet**	**xgbTree**	**glmnet**	**nnet**	**xgbTree**	**glmnet**	**nnet**
AUC for all samples	0.899	0.913	0.927	0.705	0.733	0.719	0.875	0.92	0.948
AUC for all patients	0.611	0.861	0.917	0.639	0.528	0.528	0.694	0.861	0.917
**Peripheral**									
	**Stereology**			**qPCR**			**Stereology and qPCR**		
	**xgbTree**	**glmnet**	**nnet**	**xgbTree**	**glmnet**	**nnet**	**xgbTree**	**glmnet**	**nnet**
AUC for all samples	0.903	0.938	0.924	0.681	0.694	0.681	0.819	0.938	0.979
AUC for all patients	0.861	0.917	0.861	0.556	0.556	0.556	0.778	0.917	0.917
**Central**									
	**Stereology**			**qPCR**			**Stereology and qPCR**		
	**xgbTree**	**glmnet**	**nnet**	**xgbTree**	**glmnet**	**nnet**	**xgbTree**	**glmnet**	**nnet**
AUC for all samples	0.889	0.903	0.903	0.722	0.694	0.722	0.826	0.889	0.889
AUC for all patients	0.778	0.861	0.861	0.611	0.611	0.611	0.722	0.861	0.806

Results for each combination of “placenta part + type of measurement + ML model”. The best version of the model is *nnet* model (highlighted in bold) on peripheral parts of placenta using stereological and qPCR data together. In this case, “AUC for all patients” = 0.917 (corresponding to an error for just one patient-case in a sample of 15 patients: 9 controls and 6 FGR), and “AUC for all samples” = 0.979 (corresponding to an error for only one measurement, a FGR in a set of 15 × 4 samples: 9 × 4 control samples and 6 × 4 FGR samples). A complete table of errors is given in Appendix A.

## Data Availability

All relevant data are within the paper and available upon reasonable request.

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
