# Peer review of "Integrated Placental Modelling of Histology with Gene Expression to Identify Functional Impact on Fetal Growth"

_cells, 2023, doi:10.3390/cells12071093_

Round 1

Reviewer 1 Report

The manuscript entiteled 'Integrated placental modelling of histology with gene expression to identify functional impact on fetal growth' is a very fascinating topic contributing to the very important correlation of morphological findings, gene expression and placental function.

The methods are very elegant and the results interesting.

Yet, I have few comments and questions:

1. lines 258-260: Recommendation for placental diagnostics is not to take sections from the peripheral parts of the placenta. This is contrary to your method. Please include a short explanation why you decided to do so.

2. lines 264-269: how do you explain, that FGR placentas with decreased volum of intervillous space and intermediate and terminal villi, showed no difference in diffusion barrier and diffusion capacity? This is contrary to the functionality of terminal villi, dysfunctionality of distal villous hypoplasia and measurements of diffusion distance through different gestational ages. 

3. lines 276-278: Placental IGF2, 276 SLC2A3 and SLC38A1 expression was significantly lower for FGR versus controls, regard less of sampling location ...

Could you please include an explanation?

Author Response

Reviewer 1

The manuscript entiteled 'Integrated placental modelling of histology with gene expression to identify functional impact on fetal growth' is a very fascinating topic contributing to the very important correlation of morphological findings, gene expression and placental function.

The methods are very elegant and the results interesting.

Yet, I have few comments and questions:

  1. lines 258-260: Recommendation for placental diagnostics is not to take sections from the peripheral parts of the placenta. This is contrary to your method. Please include a short explanation why you decided to do so.

We were seeking an unbiased approach to optimise sampling the placenta and used the cord insertion as the landmark feature for systematic relative sampling. This method ensured that same amount of tissue was sampled, and that tissue selection was directly related to the cord in the same way as all other samples. This approach was taken to overcome the limitations of previous placental templates that assumed a circular placenta with central cord insertions, as FGR placentas can be mis-shaped with marginal cord insertions. Please see justification for our approach in lines 90 to 100 and 131 to 137 of the revised text.

  1. lines 264-269: how do you explain, that FGR placentas with decreased volum of intervillous space and intermediate and terminal villi, showed no difference in diffusion barrier and diffusion capacity? This is contrary to the functionality of terminal villi, dysfunctionality of distal villous hypoplasia and measurements of diffusion distance through different gestational ages. 

We would like to point out that the thickness uniformity index for barrier thickness tended to be more variable in the FGR group (1.32 in controls vs 1.48 in FGR, p=0.048), and that the theoretical diffusion capacity tended to be reduced in FGR (95.29 in controls vs 70.26 in FGR, p=0.047) in Table 2, but were not considered statistically different after accounting for false discovery rate due to multiple testing. As such, our FGR placentas do show slight differences in these two measures of barrier thickness and diffusion capacity. The specific diffusion capacity was not different as it is adjusted relative to fetal weight, which is reduced is FGR.

  1. lines 276-278: Placental IGF2, 276 SLC2A3 and SLC38A1 expression was significantly lower for FGR versus controls, regard less of sampling location...

Could you please include an explanation?

As shown in Figure 2, our data showed that IGF2, SLC2A3 and SLC38A1 expression were overall reduced in both the samples taken from the central and peripheral parts of the FGR compared to control placenta. We have modified the text to explicitly state this in both the results and discussion. See lines 280 to 281 and 423 to 427.

Reviewer 2 Report

The authors have done great work for this manuscript and they did a lot of effort to perform placental stereology and to establish such a complex predictive modelling. I would like to point out, that the manuscript provides a systematic collection of placental tissue from control and FGR cases. This dataset might be of great relevance for other studies and provides a precise description of placental sampling. The novelty of this study is that the authors are the first to compare systematically the impact of placental sampling on results of different FGR markers in an appropriate way. They used predictive modelling to analyse whether systematic sampling was able to overcome heterogeneity in placental morphological and molecular features. However, the authors showed that there is no significant variability in the morphological features of the placenta within and between the peripheral and central sites of the placenta.

Minor Comment 1:

Line 274: I would like to ask the authors, to describe in more detail why specifically those growth factors and nutrient transporters have been chosen for this study and please include their function in the context of FGR.

Minor Comment 2:

Line 132: The authors start with Figure 4, please change the Figure Legend to Figure 1 and all the other subsequent figure numbers as appropriate.

Author Response

Reviewer 2

The authors have done great work for this manuscript and they did a lot of effort to perform placental stereology and to establish such a complex predictive modelling. I would like to point out, that the manuscript provides a systematic collection of placental tissue from control and FGR cases. This dataset might be of great relevance for other studies and provides a precise description of placental sampling. The novelty of this study is that the authors are the first to compare systematically the impact of placental sampling on results of different FGR markers in an appropriate way. They used predictive modelling to analyse whether systematic sampling was able to overcome heterogeneity in placental morphological and molecular features. However, the authors showed that there is no significant variability in the morphological features of the placenta within and between the peripheral and central sites of the placenta.

Minor Comment 1:

Line 274: I would like to ask the authors, to describe in more detail why specifically those growth factors and nutrient transporters have been chosen for this study and please include their function in the context of FGR.

We have added a little more details as to why these genes were selected as requested in lines 275 to 278.

Minor Comment 2:

Line 132: The authors start with Figure 4, please change the Figure Legend to Figure 1 and all the other subsequent figure numbers as appropriate.

We thank the reviewer for pointing out this error and have revised the figure numbers accordingly.